# Estimating the duration of antibody positivity and likely time of *Leptospira* infection using data from a cross-sectional serological study in Fiji

**Eleanor M. Rees**[1,2]*, **Colleen L. Lau**[3], **Mike Kama**[4,5], **Simon Reid**[3], **Rachel Lowe**[1,2,6,7], **Adam J. Kucharski**[1]

**1** Centre for Mathematical Modelling of Infectious Diseases, London School of Hygiene & Tropical Medicine, London, United Kingdom, **2** Centre on Climate Change and Planetary Health, London School of Hygiene & Tropical Medicine, London, United Kingdom, **3** School of Public Health, Faculty of Medicine, The University of Queensland, Herston, Queensland, Australia, **4** Fiji Centre for Communicable Disease Control, Suva, Fiji, **5** The University of the South Pacific, Suva, Fiji, **6** Barcelona Supercomputing Center, Barcelona, Spain, **7** Catalan Institution for Research and Advanced Studies (ICREA), Barcelona, Spain

* eleanor.rees1@lshtm.ac.uk

**Data Availability Statement:** We are unable to provide individual-level seroprevalence data and demographic data because of the potential for

## Abstract

### Background

Leptospirosis is a zoonotic disease prevalent throughout the world, but with particularly high burden in Oceania (including the Pacific Island Countries and Territories). Leptospirosis is endemic in Fiji, with outbreaks often occurring following heavy rainfall and flooding. As a result of non-specific clinical manifestation and diagnostic challenges, cases are often mis-diagnosed or under-ascertained. Furthermore, little is known about the duration of persistence of antibodies to leptospirosis, which has important clinical and epidemiological implications.

### Methodology and principal findings

Using the results from a serosurvey conducted in Fiji in 2013, we fitted serocatalytic models to estimate the duration of antibody positivity and the force of infection (FOI, the rate at which susceptible individuals acquire infection or seroconversion), whilst accounting for seroreversion. Additionally, we estimated the most likely timing of infection.

Using the reverse catalytic model, we estimated the duration of antibody persistence to be 8.33 years (4.76–12.50; assuming constant FOI) and 7.25 years (3.36–11.36; assuming time-varying FOI), which is longer than previous estimates. Using population age-structured seroprevalence data alone, we were not able to distinguish between these two models. However, by bringing in additional longitudinal data on antibody kinetics we were able to estimate the most likely time of infection, lending support to the time-varying FOI model. We found that most individuals who were antibody-positive in the 2013 serosurvey were likely to have been infected within the previous two years, and this finding is consistent with surveillance data showing high numbers of cases reported in 2012 and 2013.

breaching participant confidentiality. The communities in Fiji are very small, and individual-level data such as age, sex, and village of residence could potentially be used to identify specific persons. Instead aggregated data by five year age groups and can be found here: https://github.com/erees/leptoSerology. The full data can be requested via The University of Queensland's Human Research Ethics Committee for researchers who meet the criteria for access to confidential data. Email: humanethics@research.uq.edu.au Phone: +61 (7) 3365 3924.

**Funding:** EMR was supported by Medical Research Council (grant number MR/N013638/1). RL was supported by a Royal Society Dorothy Hodgkin Fellowship. AJK was supported by Wellcome Trust and the Royal Society (grant Number 206250/Z/17/Z). CLL was supported by an Australian National Health and Medical Research Council Fellowships (grant numbers APP 1109035 and 1193826). The funders had no role in study design, data collection and analysis, decision to publish, or preparation of the manuscript.

**Competing interests:** The authors have declared that no competing interests exist.

## Conclusions

This is the first study to use serocatalytic models to estimate the FOI and seroreversion rate for *Leptospira* infection. As well as providing an estimate for the duration of antibody positivity, we also present a novel method to estimate the most likely time of infection from seroprevalence data. These approaches can allow for richer, longitudinal information to be inferred from cross-sectional studies, and could be applied to other endemic diseases where antibody waning occurs.

## Author summary

Leptospirosis is a bacterial zoonotic disease that occurs in almost all regions of the world, with a particularly high burden of disease in Oceania. It is widely considered to be a Neglected Zoonotic Disease, and it is often mis-diagnosed and under-ascertained. Very little information exists about the persistence of antibodies to leptospirosis, which is important for understanding how long individuals may have partial protection against reinfection. In this study, we show how data collected from a large population survey of leptospirosis antibodies can be used to estimate the duration of antibody persistence. Knowledge of the duration of antibody persistence enables an estimation of the duration of immunity to re-infection, which is most likely antibody-mediated. We also estimate the rate at which susceptible individuals acquire infection (force of infection), whilst accounting for antibody waning. This provides more accurate estimates of population-wide disease burden. Finally, we show how the results from a cross-sectional population survey can be used to estimate when infections may have occurred. This is particularly useful in areas with limited surveillance. This approach could be applied to other neglected diseases for which data are limited and where antibody waning occurs.

## Introduction

Leptospirosis, a zoonotic bacterial disease, is found throughout the world, but is particularly prevalent in tropical and subtropical regions [1–3]. It is widely considered to be a Neglected Zoonotic Disease [4], with an estimated 1.03 million leptospirosis cases and 58,000 deaths reported worldwide each year [1], and the disease disproportionately affects resource-limited populations [5–8]. In humans, *Leptospira* infection produces a wide range of clinical symptoms, ranging from nonspecific febrile illness to jaundice, meningitis, and liver and renal failure [6,7,9]. Recent laboratory advances isolating novel species of the genus *Leptospira* from the environment using Next-Generation Sequencing has expanded the number of named species to 68, which includes both pathogenic and non-pathogenic species, and these have been proposed to be organised into two clades, and four subclades [10–12]. Leptospira can also be serologically classified into serogroups and serovars, and serotyping based on the heterogeneity of the surface lipopolysaccharide (LPS) has led to the identification of 25 serogroups and over 300 serovars [11,13–16]. Certain serovars are more commonly associated with particular hosts, for example *Leptospira interrogans* serovar Hardjo is frequently associated with cattle, and *Leptospira interrogans* serovar Canicola with dogs [16,17]. However, these associations are not absolute, and there is considerable heterogeneity in the dominant serovars in both animals and humans each country, even in remote islands [3].

Accurate diagnosis of leptospirosis remains a challenge, particularly in low and middle-income countries. Firstly, it requires clinicians to suspect leptospirosis, and since symptoms can resemble other more prevalent acute febrile illnesses, such as dengue fever, it is often mis-diagnosed or underdiagnosed. Secondly, the laboratory tests are not always available, and there are several limitations associated with each test [18–20]. The gold-standard test for diagnosing leptospirosis infection is the microscopic agglutination test (MAT), which has a high specificity and can distinguish between serogroups. However, this test has complex technical requirements. The enzyme-linked immunosorbent assay (ELISA) test is most commonly used in this context as it is easier to perform and is more sensitive than the MAT test during the acute phase of the illness, but it is not serogroup or serovar-specific. A summary table of the advantages and disadvantages of both tests is shown in S1 Table. Since both of these tests detect specific antibodies, it is important to consider the timing of testing in relation to onset of illness, as there needs to be sufficient time for the immune response to occur, and IgG or IgM antibodies to be detectable (from five to seven days post-infection) [19].

Immunity against *Leptospira* infection appears to be mediated by humoral responses [13,21], with the antibodies produced mainly targeting the surface-exposed leptospiral LPS. Anti-LPS antibodies appear to provide immunity to homologous serovars [22,23]. In addition, IgG and IgM antibody titres remain serologically detectable three to six years following infection [24,25]. The duration of protective immunity conferred following *Leptospira* infection is uncertain, and there is evidence that reinfection does occur [17,23,26,27]. Most commonly, reinfection occurs with a different *Leptospira* serogroup, and appears to result in a milder clinical disease. This suggests some degree of cross-reactive protective immunity [17,23]. However, severe disease following reinfection with the same serovar has been observed [27]. Current understanding of leptospirosis immunity is incomplete and there are gaps in the knowledge regarding leptospiral antibody dynamics, including the duration of antibody persistence, the relationship between antibody titre and reinfection, and the peak antibody levels that occur following infection.

A systematic review found that Oceania suffers the largest per capita leptospirosis morbidity (150.68 cases per 100,000 per year), mortality (9.61 deaths per 100,000 per year) [1], and disability-adjusted life years [28]. This may be an under-estimate of the true burden of disease, as access to testing is limited in the Pacific Islands, and cases are likely to be under-diagnosed [8,29]. This was evidenced by a large population-representative serological survey conducted in Fiji in 2013, which found that 19.2% of individuals sampled had evidence of a past infection [29], yet the total number of cases reported for the five years prior to the survey was around 1,200 [30] [with Fiji population size reported to be 884,887 in 2017 Census [31]]. Leptospirosis is endemic in Fiji and has been identified as one of the four priority climate-sensitive diseases of major public health concern [32]. In addition to endemic transmission, outbreaks of leptospirosis frequently occur, usually following flooding events [33].

Serological studies of healthy individuals have been used to study the population dynamics of leptospirosis [29]. However, these studies can be problematic to interpret because antibody levels wane, and therefore it is difficult to directly compare case data to seroprevalence. Serocatalytic models can be used to overcome these limitations, as they estimate the annual force of infection (FOI, the rate at which susceptible individuals acquire infection or seroconversion) whilst accounting for antibody waning (seroreversion), thus providing a better estimate of disease burden [34]. These models have been used previously for many other diseases, including infections such as measles and rubella which induce life-long immunity, as well as infections like malaria where immunity wanes [34–36]. The aim of this study is to use seroprevalence data from Fiji to estimate the FOI and the duration of antibody persistence in Fiji. Furthermore, this paper aims to demonstrate how serological data can be used to estimate the most

likely time of infection, providing additional information to enhance the analysis and interpretation of seroprevalence studies.

## Methods

### Ethics statement

Ethical approval for this study was granted by the London School of Hygiene and Tropical Medicine (reference number 16171) and by the Fiji National Health Research and Ethics Review Committee (reference number 2019.72.NW). Informed written or thumb-printed consent was obtained from adult participants, and informed written or thumb-printed parental/guardian consent and informed assent was obtained for child participants for the 2013 Fiji seroprevalence survey data [29]. Secondary analysis of an anonymised subset of this data was used in the present study.

### Study setting

Fiji, a nation in the South Pacific Ocean, comprises of 323 islands and is classified by the United Nations as a small island developing state [37]. The two biggest islands are Viti Levu, where most of the population resides, and Vanua Levu, and together they make up 87% of the total land area in Fiji. The population size was 837,217 in 2007 [31], and it is estimated that 90% of the population in Fiji are coastal dwellers [38]. The largest administrative units are Divisions (Central, Western, Northern and Eastern) followed by Provinces (14 in total).

### Data

**2012–2013 suspected clinical leptospirosis cases in Fiji.** We used a serum bank of 199 individuals with clinical suspected leptospirosis and positive IgM-ELISA, collected from April 2012 to November 2013 tested positive using an IgM-ELISA following an outbreak in Fiji [29,33]. MATs were conducted on serum from these patients, and 66 had detectable antibodies using MAT. The MAT tests were conducted on samples collected approximately two weeks following infection, although exact time lag between the onset of illness and testing were not known.

**2013 Fiji seroprevalence survey.** A total of 2,152 participants were included in the human serosurvey conducted in Fiji from September to December 2013 [29]. The population-representative survey included healthy community members across the Central Administrative Division (on the eastern side of Viti Levu), the Western Division (on the western side of Viti Levu), and the Northern Division (the islands of Vanua Levu and Taveuni). The age of participants ranged from 1 to 90 years (mean 33.6 years, standard deviation 19.8 years) and 45.8% were males. The presence of anti-*Leptospira* antibodies in sera collected from participants was determined using the MAT with a panel of six serovars, *Leptospira interrogans* serovars Pohnpei (serogroup Australis), Australis (serogroup Australis), Canicola (serogroup Canicola), Copenhageni (serogroup Icterohaemorrhagiae), Hardjo (serogroup Sejroe), and *Leptospira borgpetersenii* serovar Ballum (serogroup Ballum). An initial panel of 21 pathogenic serovars was used on a random selection of ~10% of the total samples. In addition, this 21 serovar panel was used on 199 *Leptospira* ELISA-positive samples collected from patients with suspected clinical leptospirosis in Fiji in 2012 and 2013. The serogroups most commonly detected in the clinical and serosurvey samples were then chosen and included in the final panel of six serovars. Further details on selection of the serovars for the MAT panel have been previously described by Lau *et al.* [29]. Samples were tested at titre dilutions from 1:50 to 1:3200, and MAT titres of ≥1:50 were defined as seropositive. A higher antibody titre dilution is usually

considered indicative of a more recent infection (i.e. MAT $\geq$1:400), whilst a lower antibody titre of a past infection. MATs were conducted at the WHO Collaborating Centre for Reference and Research on Leptospirosis in Brisbane, Australia.

Of the 2,152 individuals included within the study, 417 were seropositive to at least one serovar (19.4%). The age distribution of individuals included in the study by five-year age groups is shown in S1 Fig. A total of 351 individuals were seropositive to serovar Pohnpei (84.2%), 56 to serovar Copenhageni (13.4%), 49 to serovar Canicola (11.8%), 43 to serovar Australis (10.3%), 18 to serovar Ballum (4.3%) and three to serovar Hardjo (0.7%). Of these, 89 individuals were seropositive for more than one serovar. The ages of 12 individuals were missing, and they were excluded from the analysis. The age distribution of seropositive individuals by ten-year age group by serovar is shown in S2 Fig. The distribution of MAT titres by serovar is shown in S3 Fig.

**Lupidi point-source outbreak.**   A point source outbreak of leptospirosis occurred in Italy in 1984 that involved 18 individuals who drank water from a common source that was contaminated with infected animal urine [25]. They were followed up over a five-year period, with MAT tests conducted at five different time points.

**Serocatalytic models.**   Serocatalytic models can be used to reconstruct the annual force of infection (FOI, defined as the per capita rate at which susceptible individuals are infected each year) from cross-sectional serological surveys [34]. If an infection provides long-term immunity (e.g. measles), then we would expect seroprevalence to accumulate with time, and therefore increase with age. These dynamics can be captured using a catalytic model which assumes that susceptible individuals are infected at a given rate per year (i.e. FOI), and once infected, individuals recover and remain immune. An extension of this is the reverse catalytic model, which allows for antibody decline over time, and for previously infected individuals to become susceptible again. These simple models assume a constant FOI, however, variation in FOI with age and/or over time may lead to more complicated dynamics. Examples of different seroprevalence profiles that may be observed are shown in Fig 1.

The catalytic model follows individuals from birth and assumes that there is a life-long constant FOI ($\lambda$), which is independent of age (a) and calendar year. The rate of change in the proportion of individuals who are infected *z(a)* with age is as follows:

$$z(a) = 1 - e^{-\lambda a}$$

where $\lambda$ is the FOI and *a* is age.

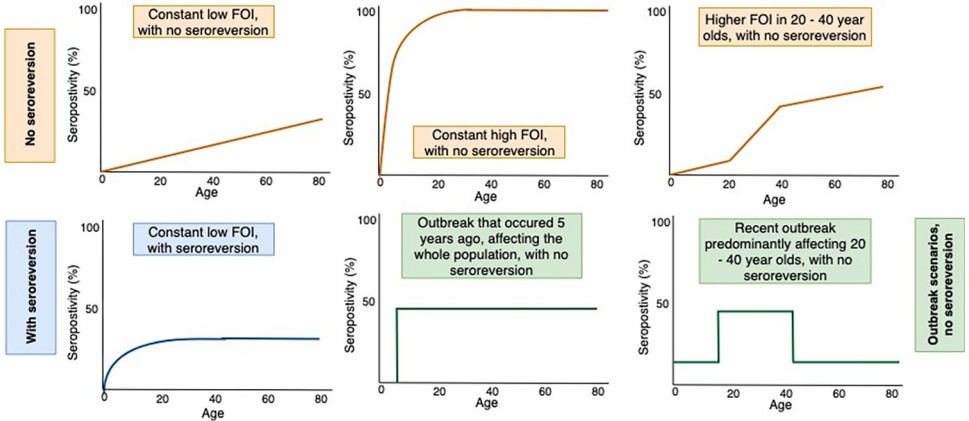

**Fig 1. Schematic representations of different possible seroprevalence profiles by age that could be observed, depending on underlying epidemic and immunological dynamics.**

The reverse catalytic model assumes that antibody prevalence declines over time, at a rate ω. The expression for the proportion of individuals aged *a* who are seropositive, *z(a)*, in the reverse catalytic model is as follows:

$$z(a) = \frac{\lambda}{\lambda + \omega}\left(1 - e^{-a(\lambda + \omega)}\right)$$

where λ is the FOI, ω is seropositivity waning rate and *a* is age. Both models assume the mortality rates for susceptible and infected individuals are the same. 1/ω is the duration of antibody persistence (years). Annual attack rates were calculated after estimating the FOI using the following expression,

$$Attack\ rate = 1 - e^{-\lambda}$$

To reflect uncertainty in knowledge of the transmission dynamics of leptospirosis in Fiji uninformative priors were chosen for the FOI and rate of waning over time. Specifically, a uniform distribution between 0 and 0.5 was chosen for the FOI (corresponding to a yearly attack rate between 0 and 39%) and a uniform distribution between 0 and 10 for the rate of waning (S2 Table).

We then fitted the reverse catalytic model by sex, administrative division and serovar. In all models, waning was held constant and FOI was allowed to vary. For the serovar-specific analyses, 89 individuals were seropositive for more than one serovar. If the titre was higher for one serovar, this serovar was used for the analyses. For 18 individuals, the titres were the same for more than one serovar, and these were labelled as "mixed". Only a small number of individuals were considered seropositive for serovar Hardjo (n = 3) and serovar Australis (n = 1), and so were excluded from the analysis. For the analysis by sex and administrative division, the same priors were used as above, a uniform distribution between 0 and 0.5 for FOI, and a uniform distribution between 0 and 10 for the rate of waning. For the analysis by serovar, the FOI was allowed to vary by serovar, whilst waning was held constant across serovars. A narrower uniform distribution between 0 and 0.1 was used for the FOI instead, whilst the rate of waning was the same (uniform between 0 and 10; S2 Table).

Waning was held constant across serovars as when the FOI is lower, FOI and waning can be more challenging to estimate. This is because there are fewer infection events over time and hence greater uncertainty. To highlight this, we did a simulation recovery study where we recovered the FOI and waning estimates from two settings, a high FOI and low FOI setting. Using the reverse catalytic model we generated two models, a high FOI model (FOI, 0.05 and waning 0.1) and low FOI model (FOI 0.005 and waning 0.1). We then sampled 50 times from each 5-year age group using a binomial distribution to generate seropositive and seronegative individuals (S4 Fig). We then re-fitted a reverse catalytic model to both datasets to estimate the FOI and waning in both settings. In the high FOI setting we were able to get similar estimates for both FOI and waning, with the input parameter estimates included within the 95% credible intervals. However, in the low FOI setting, although the true parameter values were included within the 95% credible intervals, there was much greater uncertainty in the parameter estimates (S3 Table and S5 Fig).

Bayesian inference was used to fit the serocatalytic models to empirical data, using Markov chain Monte Carlo (MCMC) with the Gibbs sampling algorithm to estimate model parameters. The models were implemented in RJags (version 4–10) [39]. The Gelman-Rubin statistic was used to evaluate MCMC convergence, and a threshold of <1.1 was chosen. The effective sample size (ESS), which is the estimated number of independent samples accounting for autocorrelations generated by the MCMC run, was checked, and an ESS >200 was used. Model

selection was based on the lowest value of the widely applicable information criterion (WAIC), which balances the goodness of fit of the model with model complexity, and therefore aims to balance the risks of overfitting and underfitting [40,41]. WAIC was estimated using the R package Loo (version 2.4.1) [42]. All analysis and calculations were performed using R version 4.1.1. All R code is available on Github (https://github.com/erees/leptoSerology).

### Time-varying FOI

The models described above assumed that the FOI was constant over time. We also considered exceptions to this assumption by exploring models which allow outbreaks to occur, where the FOI was instead given as a sum of Gaussian distributions, as described in the Rsero package [43]. The timing of the outbreak and the infection probability were estimated. A model with only one outbreak was compared with models which combined a constant FOI with an outbreak. A uniform distribution between 0 and 10 was chosen for the FOI and a uniform distribution between 0 and 10 for the rate of waning. Two different priors were tested for the timing of the outbreaks based on the earlier calculated duration of antibody persistence (S2 Table).

Analysis of time-varying FOI was performed using the Rsero package [43]. Parameter estimation was performed using MCMC using the No-U-Turn sampler (NUTS) sampling algorithm. Convergence was assessed by ensuring Gelman-Rubin statistic <1.1 and effective sample size >200. WAIC was estimated using the R package Loo (version 2.4.1) [42].

### Reconstructing timing of historic infections

Using the MAT antibody titres from the 2013 Fiji seroprevalence survey, we estimated the timing of infection of participants. Due to uncertainty associated with individual titre estimates–and hence timings–the dynamics of infection was aggregated and reported as the population level expectation. Firstly, we estimated the rate at which individual responses wane by one antibody dilution titre. This was done using data from the point source outbreak in Italy reported by Lupidi *et al.* [25]. Since leptospirosis is not endemic in Italy, this presented an opportunity to look at antibody decay, in a setting where reinfection is unlikely. Using these data, decline in antibody titres for each individual was assumed to follow exponential decay, so that the log antibody titre decays linearly with time. A linear mixed effects model was used, with a random effect for the intercept as described below:

$$\text{titre} \sim \text{time} + (1|\text{id}) + \varepsilon$$

We implemented this model in R using the lme4 package [44]. Three serovars were identified by the MAT in the Lupidi *et al.* [25] point source outbreak, however, it was not clear which was the infecting serovar (likely due to cross-reactivity of the MAT). Therefore, all three serovars were analysed individually, and the results pooled.

To reconstruct the timing of infection from the 2013 seroprevalence survey we combined the 2012 Fiji clinically suspected cases with the estimated rate at which individual responses wane by one antibody dilution titre. First, using the 2012 Fiji clinically suspected cases (n = 199), we estimated the geometric mean antibody titre from the MAT-positive cases (n = 66). We then used the MAT antibody distribution of the 2012 Fiji clinically suspected cases, combined with the antibody decay estimates from Lupidi *et al.* [25], to analyse the 2013 seroprevalence data. For each titre level from the 2013 seroprevalence survey, the possible initial titre levels were estimated, based on the proportions from the 2012 Fiji clinically suspected case distribution results. Using the rate of decline of antibody titres, the initial titre levels were transformed into an estimated time since infection, reconstructing the potential timing of infection at the population level. An example for one titre is shown in Fig 2. As individuals

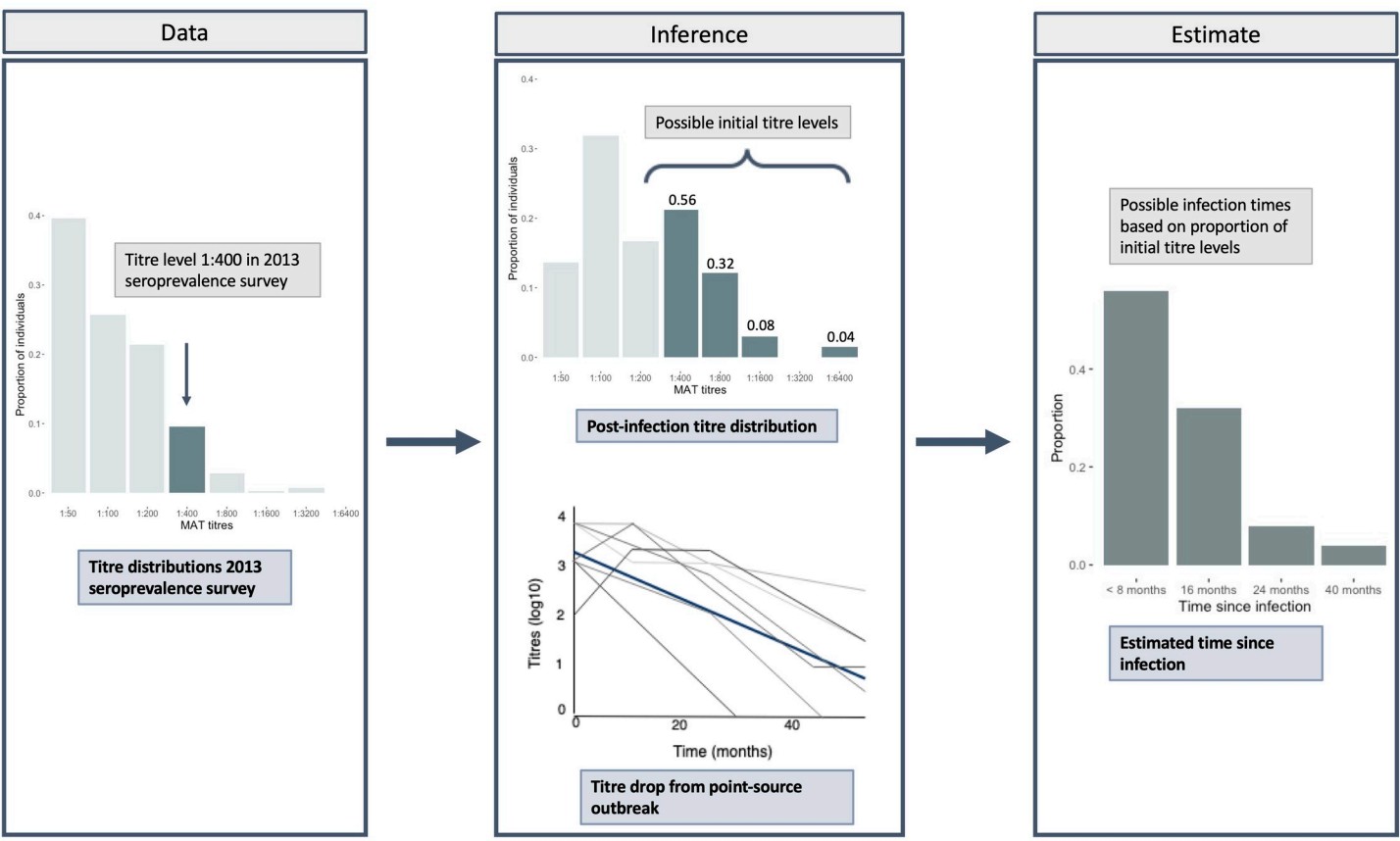

**Fig 2. Schematic representation of the methods used for estimating the historic time of infection from the seroprevalence survey data.** Firstly, we have the titre distributions from the 2013 seroprevalence survey (Data panel). Then we have the titre distribution of recent infections (Inference panel, upper plot) and the estimated antibody titre decay rate (Inference panel, lower plot). These are both used to estimate the possible time of infection based on the initial titre level (Estimate panel). As an example, individuals who had a titre level of 1:400 in the 2013 seroprevalence survey (Data panel) could have a titre level of 1:400 or higher (upper panel "Inference") ~ two weeks post-infection. If the initial titre level was higher than 1:400, the antibody titre must have waned to reach 1:400. The proportion of initial titre levels was obtained from 2012 clinically suspected cases, and in this case, 56% were likely to have had an infecting titre of 1:400 while 44% were likely to have had a higher infecting titre. Then, transforming this using the antibody decay rate from a point source outbreak in Italy (lower panel "Inference"), we can say that 56% are likely to have been infected <8 months ago (Estimate panel). This was repeated for each dilution level.

could be seropositive for more than one serovar, two separate analyses were conducted, one where infections with different serovars were assumed to be independent events (n = 520), and one where only the highest titre was used (n = 417).

Since samples were obtained approximately two weeks post-infection from recently infected individuals, we hypothesised that antibody titres may not have peaked. We compared the geometric mean antibody titres to the peak antibody titres reported in Lupidi *et al*. [25] and found a 1–3 fold difference in geometric mean antibody titres. Therefore, we conducted a sensitivity analysis where the distribution of recently infected individuals was shifted, corresponding to a higher overall geometric mean, and estimated a new distribution for time of infection.

## Results

### Serocatalytic models

When catalytic and reverse catalytic models were fitted to the 2013 Fiji seroprevalence data, we found that the reverse catalytic model, which allows for seroreversion, fitted the data better

**Table 1. Parameter estimates for the force of infection (FOI) and waning rate from the catalytic and reverse catalytic model (median [95% CrI]).**

| Model | FOI (95% CrI) | Waning rate (95% CrI) | WAIC |
|---|---|---|---|
| Catalytic Model | 0.007 (0.006–0.007) | - | 2215 |
| Reverse catalytic model | 0.032 (0.022–0.053) | 0.12 (0.08–0.21) | 2091 |

FOI, force of infection; WAIC, widely applicable information criterion; CrI, credible interval.

(Table 1; Fig 3), with a lower estimated widely applicable information criterion (WAIC difference = 124). The WAIC is an information criterion used for model selection, that aims to balance model complexity with fit to the data. The reverse catalytic model estimated the duration of antibody persistence to be 8.33 years (95% CrI: 4.76–12.50 years), and the force of infection, FOI, to be 0.032 (95% CrI: 0.022–0.053) (Table 1), which corresponds to an annual attack rate 3.15% (95% CrI: 2.18% - 5.16%).

## FOI and waning by serovar, sex and administrative division

The reverse catalytic model was also extended to explore whether sex, administrative division and serovar affected the estimated rate of seroreversion (Table 2). Overall, the estimates of seroreversion for all three models were consistent with the previous estimate of the reverse catalytic model using aggregated data. The differences in FOI estimated between the groups in the model correspond to the observed variation in seroprevalence measured in the serosurvey. When analysed by sex, a higher FOI was observed in males compared with females, and this is in accordance with the results from the serosurvey, where it was found that the seroprevalence in males was higher than females. When analysed by administrative division, the Western Division was found to have the highest FOI compared with the Central and Western Divisions, although the credible interval was large. Finally, the results by serovar were also in accordance

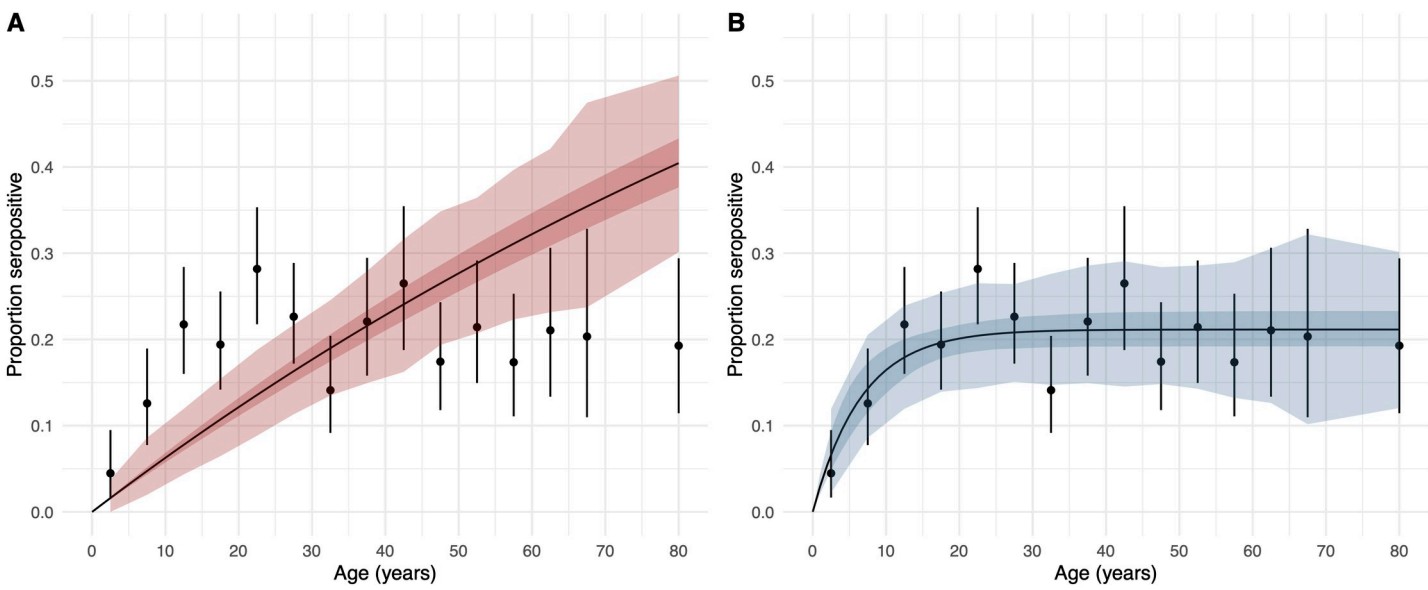

**Fig 3. Proportion of seropositive individuals by age (black points represent the mean and the error bars represent the binomial 95% confidence intervals), from national serosurvey conducted in Fiji in 2013 (n = 2,152).** Results from the catalytic model is shown in red (A) and reverse catalytic model is shown in blue (B), including model 95% credible intervals (darker shading) and the sampling uncertainty (binomial, lighter shading).

**Table 2. Parameter estimates for the FOI and waning for the reverse catalytic model by sex, by administrative division and by serovar (median [95% CrI]).**

| Model | FOI (95% CrI) | Waning (95% CrI) |
|---|---|---|
| Reverse catalytic model by sex | Female: 0.025 (0.017–0.040)<br>Male: 0.042 (0.029–0.067) | 0.120 (0.078–0.200) |
| Reverse catalytic model by administrative Division | Central: 0.033 (0.022–0.058)<br>North: 0.035 (0.022–0.065)<br>West: 0.038 (0.025–0.068) | 0.135 (0.085–0.250) |
| FOI allowed to vary by serovar, waning held constant | Ballum: 0.0009 (0.0004–0.003)<br>Canicola: 0.0028 (0.0015–0.0070)<br>Copenhageni: 0.0021 (0.0011–0.0053)<br>Pohnpei: 0.0340 (0.0208–0.0830) | 0.175 (0.101–0.449) |

FOI, force of infection; CrI, credible interval.

with the serosurvey (S6 Fig). Serovar Pohnpei was found to have the highest FOI, which was also the most commonly identified serovar in the serosurvey.

## Time-varying FOI

Our baseline catalytic and reverse catalytic models assumed a constant FOI. Therefore, we also assessed whether varying the FOI over time impacted the estimate for seroreversion (Table 3). Firstly, a constant FOI was assumed, but with the addition of one recent outbreak (allowed to occur two years prior to the seroprevalence survey). This approach was then extended, allowing for the outbreak to have occurred anytime in the five years preceding the seroprevalence survey. These models had similar estimates of seroreversion [7.25 years (3.36–11.36), for the constant FOI with one outbreak in the last five years], which were comparable with the estimate from the simple reverse catalytic model. There was little difference in WAIC between the two models which included a constant FOI and an outbreak, indicating both models performed equivalently well. Furthermore, the estimates of WAIC were similar to the reverse catalytic model (Table 3). The timing of the outbreak, when allowed to occur in the preceding five years, estimated the outbreak to be in April 2013 (95% CrI: September 2009—December 2013; S7 Fig), albeit with wide credible intervals and there was a lot of uncertainty regarding the height of the peak. Finally, a subsequent model assessed the effects of having no constant FOI and one outbreak (outbreak only scenario) occurring in the 10 years preceding the survey. This model estimated a higher rate of seroreversion, and a higher WAIC (WAIC difference: 13, compared with constant FOI with an outbreak in the previous five years), indicating that the model did not have as much support.

## Reconstructing historic time of infection

Our above modelling analysis used population level seroprevalence data to estimate the most likely timing of the outbreak. The model estimated a recent outbreak, however the credible

**Table 3. Time-varying FOI models.** Parameter estimates for the constant FOI, outbreak timing and waning for the reverse catalytic model with a constant FOI and one outbreak in the last two years, the reverse catalytic model with a constant FOI and one outbreak in the last five years, and the reverse catalytic model with no constant FOI and one outbreak in the last ten years (outbreak only model).

| Model | Constant FOI estimate (95% CrI) | Outbreak timing (95% CrI) | Waning (95% CrI) | WAIC |
|---|---|---|---|---|
| Constant FOI with 1 outbreak (2 years) | 0.036 (0.024–0.060) | 2013–05 (2012–08–2013–12) | 0.139 (0.089–0.241) | 2089 |
| Constant FOI with 1 outbreak (5 years) | 0.036 (0.022–0.061) | 2013–04 (2009–09–2013–12) | 0.138 (0.088–0.298) | 2090 |
| No constant FOI & 1 Outbreak (10 years) | - | 2009–02 (2008–05- 2010–03) | 0.492 (0.095–0.779) | 2103 |

FOI, force of infection; WAIC, widely applicable information criterion; CrI, credible interval.

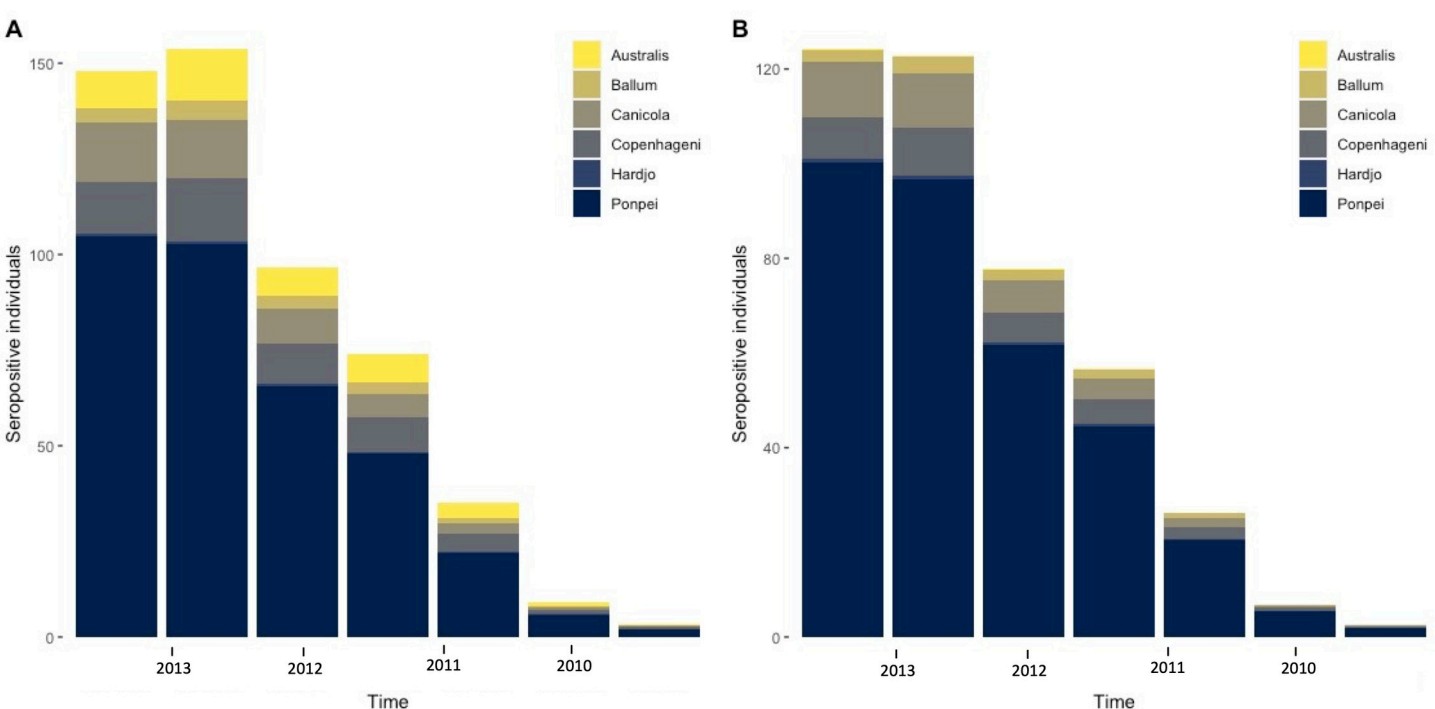

**Fig 4. Estimating the most likely time of infection from leptospirosis seroprevalence data from Fiji.** (A) assumes that individuals can be seropositive for more than one serovar at different times (n = 520), whilst (B) using results of the serovar associated with the highest titre (n = 417).

intervals surrounding the estimated timing were large. Therefore, we also conducted a complementary analysis to estimate the timing of infection at the individual level, using the MAT titres by serovar instead of aggregated binary seropositivity. First, using a mixed-effects linear model and data from the point-source outbreak from Lupidi *et al.* [25], the rate that antibody titres drop by one dilution level was estimated to be 7.92 (6.30–11.08) months. The time taken to reach undetectable levels was estimated as 6.57 years following infection (S4 Table). These results were pooled across all three serovars reported by Lupidi *et al.* [25] since there was no clear infecting serovar identified in the study. The antibody decay rate, along with the titre distribution of recently infected individuals sampled in 2012 and 2013, were then used to estimate when individuals included in the seroprevalence survey might have become infected (schematic representation shown in Fig 2). The results indicate that a recent outbreak most likely caused the majority of infections, with estimated time of infection predominantly in 2012 and 2013 (Fig 4). This was true under both assumptions of infection; firstly, where infections were assumed to be independent events, and an individual can be seropositive for more than one serovar (Fig 4A); and secondly where only the highest antibody titre was used, and we assumed individuals could not be infected with more than one serovar (Fig 4B). These results correspond with what is known from surveillance data reported by the Fiji Ministry of Health and Medical Services, which show large outbreaks in 2012 and 2013, with 563 and 453 cases reported respectively [30,33]. In comparison, an annual mean of 72 cases were reported between 2008 and 2011 (although data were known to be less accurate for 2010, where only five cases were reported). A breakdown by serovar is shown in S8 Fig.

The samples from individuals with clinically suspected leptospirosis from 2012 were collected approximately two weeks following infection. There may not have been sufficient time for antibody levels to peak, therefore the geometric mean antibody titres were compared to the

peak antibody titres from Lupidi *et al.*. The mean antibody titre in Lupidi *et al.* was found to be 1–3 dilutions higher than in the clinically suspected individuals from Fiji, so a sensitivity analysis was conducted. The 2012 titre profiles were shifted so that the mean geometric titre corresponded to those observed in Lupidi *et al.* [25]. This placed the peak of the infection further in the past, but still within the last three years (S9 Fig).

## Discussion

Serocatalytic models can be used to estimate time-dependent values such as the rate of infection and duration of seropositivity from cross-sectional seroprevalence studies [34]. They are particularly useful tools in serological studies on diseases where seroreversion occurs, as we can make comparisons between seroprevalence and surveillance data, whilst accounting for waning of antibodies, and this has important public health implications. For example, in a setting where there is a high force of infection, FOI, and rapid seroreversion, it could be wrongly concluded from an overall low seroprevalence estimate that little transmission is occurring. In our analysis, the estimated annual attack rate for *Leptospira* infection in Fiji (3.15%, 2.18% - 5.16%) using the reverse catalytic model would suggest that there may be as many as 28,000 (19,000–46,000) infections in Fiji per year, using the 2017 population census. Annually reported cases in Fiji have typically varied from a couple of hundred cases to over a thousand, but our findings quantify the potential extent of unascertained community infection. Reasons for this under-ascertainment could be due to clinical misclassification (e.g. misdiagnosis as dengue fever), limited access to laboratory diagnosis, individuals with mild symptoms not seeking health care, or asymptomatic infections [45]. While the data supports evidence that there is under-reporting, it is worth noting that the serosurvey was conducted during a period of high incidence, and the FOI may have been estimated to be lower in other years.

Using the reverse catalytic model we also estimated the persistence of detectable anti-*Leptospira* antibodies to be 8.33 years (4.76–12.50 years). Similar estimates were obtained when analysed by sex, administrative division and serovar. Furthermore, since large seasonal outbreaks of leptospirosis are known to occur in Fiji, we explored how a time-varying FOI influenced our estimates of the duration of seropositivity and found that our estimates remained similar [7.25 years (3.36–11.36), for the constant FOI with one outbreak in the last five years]. There was little difference between the WAIC estimates of the reverse catalytic model and the time-varying FOI model, indicating that both models performed equivalently using the seropositivity data. Therefore, using the seroprevalence study alone, we were not able to identify which scenario had the most support. The duration of antibody persistence estimated in this study is longer than that found by previous studies, which estimated it to be between 3–6 years [24,25]. However, the follow up duration in previous studies was between 5 and 6 years, and some individuals remained seropositive at the conclusion of the study in both Lupidi *et al.* [25] (follow up duration of five years) and in Cumberland *et al.* [24] (follow up time of six years). This indicates a longer period of follow-up may be required to accurately measure the duration of antibody persistence. We estimated (using a linear mixed effects model) that in Lupidi *et al.* the time taken to reach undetectable levels was 6.57 years, which extended beyond the follow-up period, and this is in accordance with our estimate of the duration of antibody persistence from the Fiji serosurvey, suggesting that antibody decay rates are comparable across settings. However, care needs to be taken when comparing the duration of immunity in different contexts. Fiji is an endemic setting where repeated infections are more likely. These may boost antibody responses, resulting in longer persistence of measurable antibodies [6,46]. In a different, non-endemic setting, antibody persistence may be estimated to be shorter. Therefore,

additional longitudinal datasets from settings with high prevalence would be useful to validate our results.

In our study, we focus on antibody responses that provide a correlate of *Leptospira* infection. However, understanding the dynamics of infection more fully would require more detailed analysis of the relationship between seropositivity, development of symptomatic disease and protective immunity. One of the most accurate ways to assess the duration of immunity is to conduct longitudinal reinfection studies. Reinfection generally occurs with a different infecting serovar and appears more likely to result in asymptomatic infection or mild clinical disease, suggesting protective specific-immunity but also cross-reactive protection following initial infection. However, severe disease following a second infection [17,27], and repeat infections with the same serovar have also been observed [26,27]. The exact timing of prior infection was often not known in these studies and many only had short follow-up periods, highlighting the need for prospective studies in well characterised populations with sufficient follow up periods. These studies would address many unanswered questions, including the nature and duration of immunity to *Leptospira* in terms of whether it is serovar or serogroup specific, whether it results in milder clinical disease, and finally, whether it is correlated with antibody titre levels. These questions could have implications for the successful development and deployment of a vaccine in humans.

In the absence of more detailed prospective studies, antibodies may act as a correlate for protective immunity, however, care needs to be taken in interpretation. Despite low and possibly un-detectable levels of antibodies, immunity may persist. Memory B-cells can reside outside serum and are therefore difficult to detect from blood samples, but can rapidly produce antibodies following an infection. Furthermore, immunity is not driven solely by antibody-mediated processes, as cell-mediated immunity may also play a role [13,14]. Therefore, antibody titres may under-estimate immunity against pathogens. Leptospires are extracellular pathogens, and as such humoral-mediated immunity is thought to play a central role [13,21]. Previous studies have shown that protective immunity can be transferred via the serum [47,48], demonstrating the role of antibodies, and suggesting that immunity to leptospirosis is driven primarily via the humoral immune response. Therefore, the duration of antibody persistence is likely to be a good correlate for immunity.

Since antibodies can act as a marker of exposure to infection, we explored two complementary approaches to estimate the timing of infection at the population level. In the first, we used the population seroprevalence data, and allowed for a time-varying FOI, which inferred that there was endemic transmission occurring, and a large outbreak in 2013. However, there was a lot of uncertainty regarding the timing and the size of the peak, with large credible intervals, when only the binary seroprevalence data was used. In the second approach, we used the MAT antibody titres by serovar, MAT antibody titres from clinically suspected leptospirosis cases and longitudinal information on antibody decay rates from Lupidi *et al.* [25] to estimate the most likely timing of infection, fully utilising the available seroprevalence data. From this, we found that most individuals included in the 2013 seroprevalence study were likely to have had a recent infection within the last two years. These results appear to correspond with what is known from surveillance data collected by the Fiji Ministry of Health, which show high numbers of cases in 2012 and 2013 [30,33]. We demonstrate that by incorporating additional sources of data, including longitudinal information on antibody kinetics, we were able to identify the timing of infection. Identifying a time window when infection may have occurred could be useful when analysing results from serosurveys, as this would allow for data to be chosen based on temporal proximity to the likely infection period. This may increase the accuracy of analyses and reduce confounding that may occur through the combination of disparate datasets. We did not observe any patterns of infection by serovar, suggesting that there may be

simultaneous circulation of multiple serovars in Fiji, rather than multiple outbreaks with different serovars. A previous study describing the human serosurvey in Fiji found that there were differences in serovar distribution by age and location, suggesting that there are different risk factors of disease transmission between sub-groups [29]. For example, livestock could be more important drivers in rural areas, with rodents being more important in urban areas. However, in this setting there was one dominant serovar (serovar Pohnpei), limiting the ability to identify serovar-specific risk factors. Since *Leptospira* exposure does not induce lifelong immunity (i.e antibodies wane following *Leptospira* infection), it was not possible to estimate infection beyond the time it takes for seropositivity to wane. Therefore, such data cannot provide insights further back in time than the duration of antibody waning.

The results from our analysis are to some extent limited by the quality of the data available. We used MAT titres from recently infected individuals from Fiji, however, the exact timing of infection was not known, and it is possible that individuals may not have reached their peak antibody titre levels yet [18]. In addition, standardisation of the MAT test is challenging, and the results may not be fully comparable across settings [46]. Finally, very little data exist on antibody profiles following infection with leptospirosis, and the available evidence demonstrates high levels of inter-individual heterogeneity. This is highlighted by Lupidi *et al.* [25], who reported a point source outbreak in Italy, where leptospirosis is not endemic. Each individual followed up over time in their study showed distinct antibody profiles. Despite these limitations in data quality and uncertainties in antibody dynamics, our methods were able to identify a time window in which transmission was most likely to have occurred, and which corresponds to known outbreaks in Fiji. This provides a novel way of using seroprevalence data to gain longitudinal information and insight into more recent transmission dynamics. A better understanding of antibody waning, and antibody profiles following infection, particularly given the level of inter-individual heterogeneity, would allow for this method to be further developed for leptospirosis and also other diseases.

By using serocatalytic models, we showed that it is possible to obtain insights into the underlying dynamics of leptospirosis transmission from cross-sectional data as well as providing an estimate for the duration of seropositivity. We also provide a novel method for extrapolating seroprevalence data to estimate when individuals may have become infected, showing how evidence synthesis can allow for richer, longitudinal information to be inferred from cross-sectional studies.

## Supporting information

**S1 Table. Summary, advantages and disadvantages of MAT and ELISA test used for the diagnosis of leptospirosis.**
(PDF)

**S2 Table. Description of the different models fitted and priors used.**
(PDF)

**S3 Table. Simulation recovery study.** Estimating the FOI and waning from a high FOI and low FOI setting.
(PDF)

**S4 Table. Results from the mixed-effects linear model from the point source outbreak in Italy (Lupidi *et al.*).** Antibody drop time was defined as the time taken in months for antibodies to drop one antibody titre level (e.g. from 1:100 to 1:50).
(PDF)

**S1 Fig. Number of individuals included within the 2013 leptospirosis serosurvey by five-year age groups.**
(TIFF)

**S2 Fig. Number of individuals seropositive by serovar by ten-year age groups (n = 399).** Individuals that had the same titre for two serovars, and therefore infecting titre could not be assumed, were excluded (n = 18).
(TIFF)

**S3 Fig. Distribution of MAT titres by serovar for seropositive individuals.** 89 individuals had titres for more than one serovar, and so are included more than once in this plot (n = 520).
(TIFF)

**S4 Fig. Simulation recovery study.** Sample estimates (mean and 95% binomial confidence interval) and model fit (solid line) for the high FOI (shown in orange) and low FOI (shown in blue) scenario. Under the high FOI scenario, the parameter estimates obtained were similar to the true parameter values. Under the low FOI scenario, the model was able to reproduce the data, but there was much greater uncertainty in the true underlying parameters.
(TIFF)

**S5 Fig. Simulation recovery study.** Posterior distributions for waning and force of infection (FOI) for the high FOI scenario (orange) and low FOI scenario (blue). Under the high FOI scenario, the parameter estimates obtained were similar to the true parameter values. Under the low FOI scenario, although the true parameter values were included within the 95% credible intervals, there was much greater uncertainty in the estimates.
(TIFF)

**S6 Fig.** Proportion of seropositive individuals by age (black points represent the mean and the error bars represent the binomial 95% confidence intervals), from national serosurvey conducted in Fiji in 2013 (n = 2,152) by serovar Pohnpei (A), Canciola (B), Copenhageni (C) and Ballum (D). The reverse catalytic model is shown for each serovar including model 95% credible intervals (red shading).
(TIFF)

**S7 Fig. Time-varying FOI from the constant FOI model with one outbreak (occurring in the preceding five years).**
(TIFF)

**S8 Fig. Estimating the most likely time of infection from leptospirosis seroprevalence data from Fiji by serovar.** (A) assumes that individuals can be seropositive for more than one serovar at different times (n = 520), whilst (B) using results of the serovar associated with the highest titre (n = 417).
(TIFF)

**S9 Fig. Sensitivity analysis for estimating the most likely time of infection from the seroprevalence data, using different initial titre distributions based on the geometric mean reported in Lupidi *et al*.** The initial titre distributions were shifted to correspond to a geometric mean (a) one dilution titre higher, (b) two dilutions titres higher and (c) three dilution titres higher.
(TIFF)

## Author Contributions

**Conceptualization:** Eleanor M. Rees, Adam J. Kucharski.

**Formal analysis:** Eleanor M. Rees.

**Methodology:** Eleanor M. Rees, Colleen L. Lau, Rachel Lowe, Adam J. Kucharski.

**Supervision:** Colleen L. Lau, Rachel Lowe, Adam J. Kucharski.

**Visualization:** Eleanor M. Rees.

**Writing – original draft:** Eleanor M. Rees.

**Writing – review & editing:** Eleanor M. Rees, Colleen L. Lau, Mike Kama, Simon Reid, Rachel Lowe, Adam J. Kucharski.

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
