## [Decision Letter · Decision Letter 0]

22 Feb 2022

Dear Dr Rees,

Thank you very much for submitting your manuscript "Estimating the duration of antibody positivity and likely time of Leptospira infection using data from a cross-sectional serological study in Fiji" for consideration at PLOS Neglected Tropical Diseases. As with all papers reviewed by the journal, your manuscript was reviewed by members of the editorial board and by several independent reviewers. The reviewers appreciated the attention to an important topic. Based on the reviews, we are likely to accept this manuscript for publication, providing that you modify the manuscript according to the review recommendations. 

All three Reviewers appreciated the novelty of your approaches, as well as the quality of writing in this manuscript. However, some issues need to be resolved before publication.

Sincerely,

Andre Alex Grassmann, PhD

Guest Editor

Stuart Blacksell

Deputy Editor

All three Reviewers appreciated the novelty of your approaches, as well as the quality of writing in this manuscript. However, some issues need to be resolved before publication.

Reviewer's Responses to Questions

**Key Review Criteria Required for Acceptance?**

**Methods**

-Are the objectives of the study clearly articulated with a clear testable hypothesis stated?

-Is the study design appropriate to address the stated objectives?

-Is the population clearly described and appropriate for the hypothesis being tested?

-Is the sample size sufficient to ensure adequate power to address the hypothesis being tested?

-Were correct statistical analysis used to support conclusions?

-Are there concerns about ethical or regulatory requirements being met?

Reviewer #1: Yes

Reviewer #2: I think this is a great paper. It brings together an innovative mix of methods (surveys, assays, statistical and mathematical models) to address important and understudied questions related to the epidemiology of leptospirosis.

Major comments

My major comment is that there is a potential contradiction between how the results from the applied approaches are interpreted. The reversible catalytic model makes an assumption of historically constant transmission, and was used to provide two of the key estimates: the duration of antibody persistence = 8.33 (4.76, 12.5) years; and the annual attack rate = 3.15% (2.18%, 5.16%). In contrast, two different approaches: (i) time-dependent sero-catalytic models; and (ii) longitudinal antibody kinetic analysis suggest that the majority of transmission was recent. 

The authors should weight the evidence for the two competing scenarios (constant transmission or recent outbreak), and then highlight the results from the scenario with best evidence. Retaining the other estimates as supplementary information.

Minor comments

I really like the use of the Italian outbreak for estimation of time since infection. I think there are two important limitations that should be noted. Firstly, the Italians had no pre-existing immunity. In contrast the Fijians may have had pre-existing immunity, so they would have started at a higher baseline, potentially causing the estimated times since infection to be biased towards being more recent. A second important point is on the comparability of the assays used for the Italian and Fijian data. I’m not an expert in MAT assays, so can’t judge the details, but my understanding is that being a more functional assay, comparability tends to be good. No need for a major overhaul, an acknowledgement of limitations in the discussion should be sufficient.

A nice point which I think could be emphasized a bit more is the consistency between the estimated duration of sero-positivity from the Italian data and the estimate from the sero-catalytic model. In general, I would put more weight on estimates of duration of immunity from longitudinal datasets than estimates from cross-sectional data.

“In our analysis, the estimated annual attack rate for Leptospira infection in Fiji (3.15%, 2.18% - 5.16%) would suggest that there may be as many as 28,000 infections in Fiji per year, using the 2017 census”

I think uncertainty should be propagated into burden estimates.

In Supplementary Figure 1, the recent FOI looks very high.

Data and code should be made open access. The GitHub link provided did not work for me.

Reviewer #3: The methods are well described, with sufficient data for the mathematical analysis. All applied analysis are well suited for reaching the proposed objectives.

**Results**

-Does the analysis presented match the analysis plan?

-Are the results clearly and completely presented?

-Are the figures (Tables, Images) of sufficient quality for clarity?

Reviewer #1: Yes

Reviewer #2: (No Response)

Reviewer #3: The results are clearly presented and discussed, with implications well explained. Figures and Tables are in sufficient quality.

**Conclusions**

-Are the conclusions supported by the data presented?

-Are the limitations of analysis clearly described?

-Do the authors discuss how these data can be helpful to advance our understanding of the topic under study?

-Is public health relevance addressed?

Reviewer #1: Yes

Reviewer #2: (No Response)

Reviewer #3: Conclusions are well fitted to the presented results, and the author are critical about the limited quality of the data available. Results and conclusion were used to address the epidemiological situation in Fiji, and the author proposed that this model could be applied to infer force of infection and probable time of infection in different contexts.

**Editorial and Data Presentation Modifications?**

Reviewer #1: (No Response)

Reviewer #2: (No Response)

Reviewer #3: (No Response)

**Summary and General Comments**

Reviewer #1: Overall, I thought the paper was great! 

# Introduction

The introduction is very well written. 

# Methods

I think a brief phrase on the rationale for the selection of these serovars would be useful. The authors reference another paper, but just a quick summary so readers don't have to jump into the other reference might be nice. 

For the reverse catalytic models that were serovar specific, was the rate of waning estimated independently for each? It appears in the results that waning was assumed to be the same across the serovars? It would be interesting to see if these emerge consistently in the separate models. In some ways, I think that these results should really be analyzed in a serovar specific way, given the biology of lepto (although the authors themselves discuss the limitations of the mat given cross-reactivity and also fuzziness biologically of serovars). The authors note that they use a narrower prior on the FOI for the serovar specific analyses, which maybe indicates that these data are not robust enough for serovar specific estimates of foi and waning, which I think is totally fine, but it would be good to present these results at least in the supplementary materials and state so up front. I think this is less of an issue as the subsequent analyses at the individual level largely help with this limitation. 

The link to the github repository was broken?

#Results

A couple of visualization questions/suggestions:

In figure 3, how are the raw data being presented? Are the black dots with the error bars proportion of indivduals that are seropositive and the binomial uncertainty around that proportion? How wide are the age bins? It would be nice to know how many total individuals are in each of those bins. A supplementary figure showing the age distribution would be nice and also the age distribution of seropositives by each serovar. It would also be nice to see the same type of figure in Fig 3B for each of the serotypes. 

For figure 4 and it's counterparts in the supplement, a couple of suggestions:

- Reverse the axis so that it goes from earliest to latest (or in this case the time of sampling). I think this is a bit more intuitive for the reader. 

- I personally find stacked bar graphs really hard to see. Could the serovars be split into separate panels? Or density plots if there are enough data?

It would also be nice to see how much uncertainty there is for each individual estimate of timing of infection (maybe just an x-y plot with titre on x and jittered points showing each individual estimate of the time from initial infection with uncertainty on the y) for each serovar? I may be misinterpreting the output of the model, but this seems important for evaluating whether this individual method works. 

# Discussion

Very nice discussion, the authors discuss many of the challenges in analyzing lepto data that are relevant to understanding their results. A bit on the specific serotypes in this study and the host biology would be nice (unless I just missed it).

Reviewer #2: (No Response)

Reviewer #3: Introduction

The number of species is outdated, since the WGS approach and ANI cutoff of 95% proposed by Vincent et al., (2019) currently recognizes 64 species divided in four subclades. Latter, Korba et al. (2021) isolated and identified additional 2 P1 (pathogenic) and 2 S1 (saprophytes) species. 

When the authors stated that ELISA protocols detect antibodies against both pathogenic and saprophytic species, to which antigens are they refereeing to? There are several recombinant proteins used as antigens that are specific to pathogenic species, as LipL32, OmpL1, Lig proteins, etc…Please, clarify or expand this statement.

Methods

The 199 individuals who had clinical suspected leptospirosis were all positive by IgM ELISA? This excerpt is a little ambiguous. 

Can the MAT titers of the 66 patients that were infected in a very close timeframe be used to correlates the MAT titers to distinct serovars to infer they intrinsic immunogenicity (more of that in the general comments).

Was antibody waning rate also calculated by Lupidi et al in their work? 

General comments

As immune response is multifactorial, reflecting host age and immunocompetence, and bacterial inoculum size and serovar/species, can MAT titer be generically used to infer the probable time of infection. We observed a very heterogeneous MAT titers among patients in very similar timeframes. 

Also, is there any data on how MAT titers of individuals infected with serovar Pohnpei correlates to those with serovar Copenhageni or other serovars? 

Could this serological survey also be done by IgM/IgG ELISA instead of MAT? Do the authors think that the seroprevalence would be higher? Higher IgM would be indicative of more recent infections.

PLOS authors have the option to publish the peer review history of their article (what does this mean?). If published, this will include your full peer review and any attached files.

Reviewer #1: No

Reviewer #2: No

Reviewer #3: Yes: Luis Guilherme Virgilio Fernandes

Figure Files:

Data Requirements:

Reproducibility:

References

---

## [Decision Letter · Decision Letter 1]

17 May 2022

Dear Ms Rees,

We are pleased to inform you that your manuscript 'Estimating the duration of antibody positivity and likely time of Leptospira infection using data from a cross-sectional serological study in Fiji' has been provisionally accepted for publication in PLOS Neglected Tropical Diseases.

Best regards,

Andre Alex Grassmann, PhD

Guest Editor

Stuart Blacksell

Deputy Editor

Reviewer's Responses to Questions

**Key Review Criteria Required for Acceptance?**

**Methods**

-Are the objectives of the study clearly articulated with a clear testable hypothesis stated?

-Is the study design appropriate to address the stated objectives?

-Is the population clearly described and appropriate for the hypothesis being tested?

-Is the sample size sufficient to ensure adequate power to address the hypothesis being tested?

-Were correct statistical analysis used to support conclusions?

-Are there concerns about ethical or regulatory requirements being met?

Reviewer #2: I wish to thank the authors for their detailed responses to the comments in my review, and the associated changes made to the manuscript. I believe the manuscript has been substantially improved and I'm happy to recommend it be accepted.

Michael White

Reviewer #3: (No Response)

**Results**

-Does the analysis presented match the analysis plan?

-Are the results clearly and completely presented?

-Are the figures (Tables, Images) of sufficient quality for clarity?

Reviewer #2: (No Response)

Reviewer #3: (No Response)

**Conclusions**

-Are the conclusions supported by the data presented?

-Are the limitations of analysis clearly described?

-Do the authors discuss how these data can be helpful to advance our understanding of the topic under study?

-Is public health relevance addressed?

Reviewer #2: (No Response)

Reviewer #3: (No Response)

**Editorial and Data Presentation Modifications?**

Reviewer #2: (No Response)

Reviewer #3: (No Response)

**Summary and General Comments**

Reviewer #2: (No Response)

Reviewer #3: (No Response)

PLOS authors have the option to publish the peer review history of their article (what does this mean?). If published, this will include your full peer review and any attached files.

Reviewer #2: **Yes: **Michael White

Reviewer #3: **Yes: **Luis G V Fernandes

---

## [Editor Report · Acceptance letter]

2 Jun 2022

Dear Ms Rees,

We are delighted to inform you that your manuscript, "Estimating the duration of antibody positivity and likely time of *Leptospira* infection using data from a cross-sectional serological study in Fiji," has been formally accepted for publication in PLOS Neglected Tropical Diseases.

Best regards,

Shaden Kamhawi

co-Editor-in-Chief

Paul Brindley

co-Editor-in-Chief
